# Numerical/Experimental Study of Process Optimization Conducted on an Al-Cu Alloy Produced by Combined Fields of Applied Pressure and Ultrasonic Vibration

Jinxue Wang [1], Khashayar Khanlari [2], Yali Chen [3], Bo Lin [1,3,*], Yang Zhang [4] and Weiwen Zhang [4]

1   Guizhou Suwei Materials Research Institute Limited Company, Guiyang 550025, China; vilowang@163.com
2   Département de Génie Chimique et de Génie Biotechnologique, Université de Sherbrooke, 2500 Boulevard de l'Université, Sherbrooke, QC J1K 2R1, Canada; khashayar.khanlari@usherbrooke.ca
3   School of Mechanical Engineering, Guizhou University, Guiyang 550025, China; 18798438410@163.com
4   School of Mechanical and Automotive Engineering, South China University of Technology, Guangzhou 510640, China; 13560195121@163.com (Y.Z.); mewzhang@scut.edu.cn (W.Z.)
*   Correspondence: linbo1234@126.com

**Abstract:** Single-pressure and ultrasonic action have their own unique advantages in the treatment of metal melts. When the two are combined into a composite field, the advantages of a single physical field can be fully utilized. So, the cavitation and acoustic streaming effects characteristics of an Al-5.0Cu alloy treated under different coupling process parameters, related to applied extrusion pressure and ultrasonic vibration, were analyzed by combining numerical simulation and experimental verification. The simulation results were experimentally verified by quantitative analysis of the microstructure, the melt, and its macro characteristics. The results show that the closing time t decreases with an increase in the extrusion pressure. In addition, when the ultrasonic power and extrusion force are increased simultaneously (100 MPa and 1 kW), the average grain size and the proportion of columnar grains reach the ideal effect. The influence of pressure parameters is greater, which will also lead to an increase in the proportion of columnar crystals. By optimizing the parameters, the grain size can be further reduced, the proportion of columnar crystal structure can be reduced, and fine and uniform equiaxed crystal structures can be easily obtained.

**Keywords:** Al-Cu alloy; ultrasonic-pressure coupling field; numerical simulation; process optimization

## 1. Introduction

The aluminum alloy processing industry is one of the sectors with the highest social contributions in China's non-ferrous metal industry. However, a majority of high-performance aluminum alloys in China still rely on imports [1], especially the Al-Cu alloy. At present, domestic research on the preparation of high-strength and toughness, large-size, and uniform-composition aluminum alloy ingots is lower than the world's advanced level. In addition, as an important structural material, the Al-Cu alloy is widely used in aviation, automobile, and aerospace fields [2–4].

Numerous studies have found that the application of external fields such as pressure fields or ultrasonic vibrations during the solidification of aluminum melts can substantially improve the microstructure of castings, thereby enhancing the mechanical properties of aluminum alloys [5–7]. Currently, power ultrasound has been widely used in the processing of non-ferrous metals, iron, and steel metallurgy. By applying ultrasonic vibration before or during solidification, the solidification structure can be improved and the mechanical properties can be enhanced. However, the limitations of power ultrasound are twofold. On one hand, it generates localized overheating effects, and on the other hand, it exhibits acoustic attenuation characteristics [8]. Therefore, the effective range of ultrasonic vibration is only in the area near the end face of the horn, which seriously restricts the

effect of power ultrasound. Squeeze casting combines the advantages of both the die forging and die casting processes, making it a novel and widely recognized special casting technique. Wu Shusen's [9] group prepared a new high-strength and tough aluminum alloy by applying ultrasonic, followed by the extrusion casting process. The research findings demonstrate that the mechanical properties of the alloy were greatly improved compared to the ones that were solely treated by the ultrasonic or processed by the squeeze casting process. Li et al. [10] adopted an ultrasound followed by a semi-solid state rheological process to obtain a high Fe-content Al-Si alloy and found that the corrosion resistance of the alloy was significantly improved. This was mainly attributed to the refinement effects of ultrasonic and pressure fields on the microstructure of parts. For the microstructural and mechanical properties of aluminum alloys treated under the action of physically coupled fields, some studies have also been carried out by different researchers in recent years [11,12]. Zhang's group [11] studied the effects of ultrasonic power and casting speed on the solidification structure of semi-continuous casting 7050 aluminum alloy ingots. They found that the microstructure can be refined and the casting defects can be reduced when the ultrasonic power is 240 W, so as to improve the mechanical properties of the alloy. Haghayegh et al. [12] found that the grain size can be further refined through electromagnetic–ultrasound coupling, resulting in an improvement in the tensile strength, yield strength, and elongation of the cast ingot. Consequently, the combination of ultrasonic and applied pressure coupling fields is expected to result in obtaining uniform and fine aluminum alloy structures. This is in accordance with the discussed purpose of strengthening and toughening aluminum alloys. Nevertheless, there is still a lack of systematic research and understanding on the coupling effect mechanisms and process parameter optimization of the parts treated simultaneously by the ultrasound and high-pressure fields.

Therefore, a new ultrasonic–pressure coupling casting process is proposed in this paper. By fully leveraging the respective advantages of both and through the interactive coupling between them, the localized deficiencies of each individual physical field can be compensated. As a result, a cast ingot with fine and uniform tissue, free from defects, can be prepared, further improving the macro and microstructure. Fluent(19r3), a computational fluid dynamics software, was employed to numerically simulate the cavitation effect and acoustic velocity in the Al-Cu melt treated under ultrasound–pressure coupling. The effects of the coupling process of the ultrasound and high-pressure fields on the volume fraction of the cavitation bubbles, pressure distribution, and acoustic velocity within the melt were analyzed. The coupling field mechanisms on the aluminum melt were clarified, and the process parameters of ultrasound power and static pressure were optimized to obtain the desired fine microstructure in the processed parts. These were performed along with the corresponding experimental verifications.

## 2. Mathematical Model and Solution of Multi-Field Coupling
### 2.1. Simulation and Control Equations

In this paper, the established model is a two-dimensional model where the velocity of the ultrasound during propagation is assumed to be temperature-independent. In addition, the melt was considered a non-Newtonian fluid consisting of both liquid and vapor phases, and it was assumed that a small amount of non-condensable gas was generated during the process. The control equations for the numerical simulations are shown below:

According to the sound pressure wave equation, the sound pressure satisfies the Helmholtz equation [13]:

$$\nabla^2 - \frac{\omega^2}{C^2} P = 0 \tag{1}$$

The variation of the liquid bubble radius with time under ambient pressure is as follows:

$$R\ddot{R} + \frac{3}{2}\dot{R}^2 = \frac{P_h}{\rho_l} - F(t) \tag{2}$$

The model considered various factors such as surface tension, viscosity coefficients, and other relevant parameters. The Rayleigh–Plesset equation was obtained by modifying the above equation [13]:

$$R\ddot{R} + \frac{3}{2}\dot{R}^2 = \frac{1}{\rho_l}\left(P_0 - P_V - \frac{2\sigma}{R} - \frac{4\mu}{R}\dot{R}\right) \tag{3}$$

where $P$ is the sound pressure (Pa), $\omega$ is the angular frequency (rad/s), $c$ denotes the speed of sound (m/s), $R$ refers to the bubble radius (μm), $P_0$ represents the initial pressure of the bubble ($1.013 \times 105$ Pa), $P_v$ signifies the vapor pressure, $P_h$ is the ambient pressure around the bubble, $\rho_l$ represents the liquid density (kg/m$^3$), $\sigma$ stands for the surface tension (N/m), $\mu$ denotes the fluid viscosity ($1.12 \times 10^{-3}$ Pa·s), and $F(t)$ refers to the sum of the surface and mass forces acting on the bubble (N).

In the cavitation model, the melt flow was commonly assumed to be a homogeneous mixture of gas and liquid phases, and the concept of the volume fraction was introduced to define the mixture density $\rho_m$. By substituting the mixture density ($\rho_m$) into Equation (5), the equation of motion for the gas volume fraction was obtained.

$$\rho_m = \alpha_v\rho_v + (1 - \alpha_v)\rho_l \tag{4}$$

$$\frac{\partial(\alpha_v\rho_m)}{\partial t} + \nabla(\alpha_v\rho_m u) = S_e - S_c \tag{5}$$

$$S_e = C_e\frac{v_{ch}}{\sigma}\rho_l\rho_v\sqrt{\frac{2(P_{sat} - P)}{3\rho_l}}(1 - f) \tag{6}$$

$$S_c = C_c\frac{v_{ch}}{\sigma}\rho_l\rho_v\sqrt{\frac{2(P_{sat} - P)}{3\rho_l}}f \tag{7}$$

where $\alpha_v$—the volume fraction of steam. $u$—the steam velocity (m/s). $f$—the gas mass fraction in the cavitation mixture flow. $S_e$—the steam generation source term. $S_c$—the source term for vapour condensation. $v_{ch}$—characteristic velocity (m/s), and the intensity of the local turbulence is positively related to its value. $P_{sat}$—liquid-phase saturated vapor pressure (Pa) [14]. $C_e$, $C_c$—model constants, which were set to 0.02 and 0.01, respectively, based on the relevant literature references (model constants: 0.02 and 0.01 from the literature, respectively) [13,15–18].

The relationship between the volume component $\alpha$ of the gas in the mixed flow and the radius of the individual bubbles could be further deduced as follows:

$$\alpha = N\frac{4}{3}\pi R^3 \tag{8}$$

$N$ is a constant in the calculation process; here, it was taken as $10^{13}$.

Therefore, the gas mass conservation equation in the cavitation mixture flow could be derived as follows:

$$\frac{\partial}{\partial t}(\rho_m f) + \nabla(\rho_m u f) = \nabla(\gamma\nabla f) + S_e - S_c \tag{9}$$

The cavitation model equation was finally calculated from the Rayleigh–Plesset equation, as shown below [13]:

$$\rho_m\left[R\frac{d^2R}{dt^2} + \frac{3}{2}\left(\frac{dR}{dt}\right)^2\right] = \left(P_0 - P_v + \frac{2\sigma}{R}\right)\left(\frac{R_0}{R}\right)^{3b} - \frac{2\sigma}{R} - \frac{4\mu}{R}\left(\frac{dR}{dt}\right) - P_0 + P_v + P_m sin\omega t \tag{10}$$

In which $\gamma$ refers to the effective exchange coefficient, $b$ is set to 3/4, and $P_m$ represents the acoustic pressure amplitude.

## 2.2. Geometric Model Setting

### 2.2.1. Geometric Model

This study focuses on investigating the cavitation effects in the Al-5.0Cu alloy melt. It also aims to explore the differences in the internal flow of the melt under ultrasonic-pressure coupling. Due to the short period of cavitation bubble growth–collapse, the phase transition process of melt during nucleation and the growth stages were ignored. thus preventing excessive momentum and heat source term generation. The finite element model of the ultrasonic–pressure coupling is shown in Figure 1. Because of the large size of the computational model, a hybrid meshing method is employed to enhance computational efficiency and save computation time. The mesh of the whole coupling calculation area is triangular mesh, and the mesh size is 0.08 mm. In addition, since the cavitation effect primarily occurs at the end face of the variable amplitude rod, it was necessary to refine the mesh locally. Therefore, compared to the rest of the mesh with a size of 0.08 mm, the mesh size in that region was reduced to 0.04 mm.

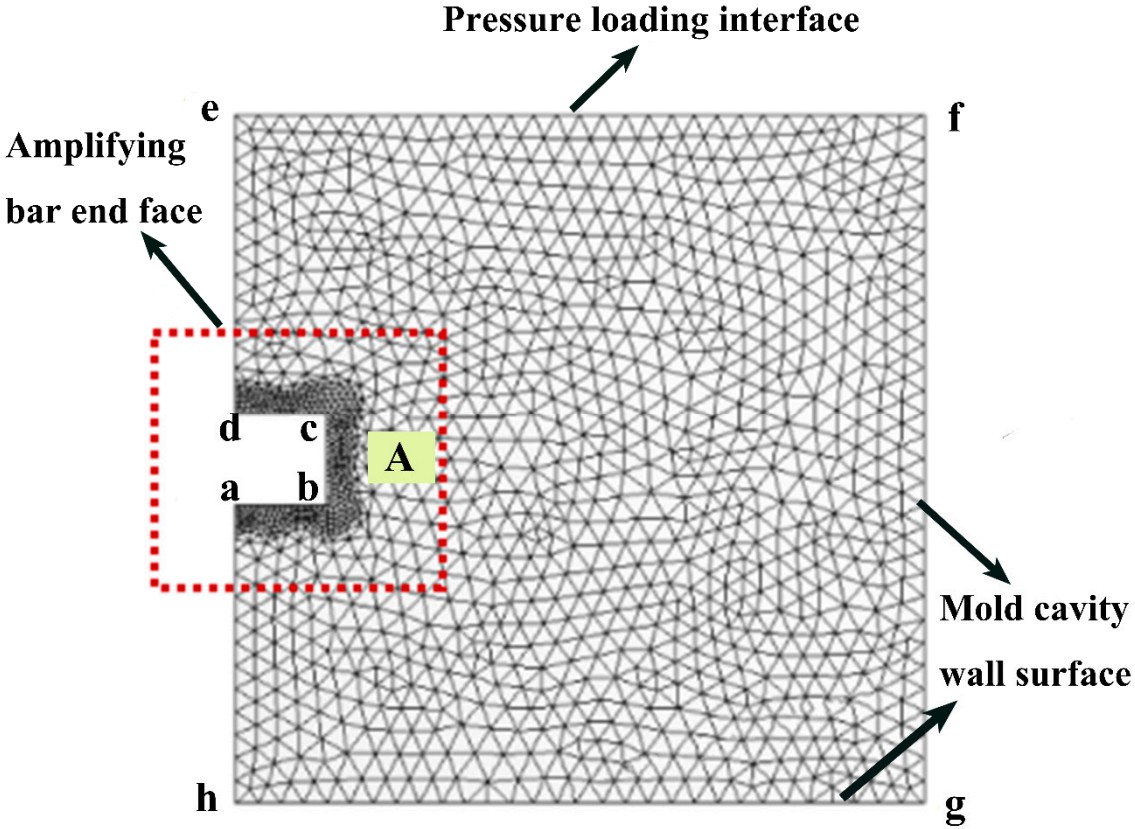

**Figure 1.** Boundary and mesh division of ultrasonic-pressure coupling calculation.

The boundaries of each wall surface in the model have no-slip conditions, and the initial pressure at the pressure inlet was set to zero. Additionally, the contact part between the ultrasonic variable amplitude rod and the melt was treated as a moving boundary. The casting temperature in the experiment was 710 °C, and the initial temperature of the mold and melt is displayed in Figure 2. Ultrasound vibration with a frequency of 20 kHz possesses moderate penetration depth, high mechanical power, and a low cost of production, making it an ideal choice in many application fields [19]. The ultrasonic device operated at a vibration frequency of 20 kHz with a variable amplitude rod of A = 10 μm. In the model, the dimensions were as follows: ab = 10 mm, cd = 10 mm, ef = 60 mm, and ha = 25 mm. The numerical solution was performed with a time step of $1 \times 10^{-4}$ s for a total solving time of 120 s.

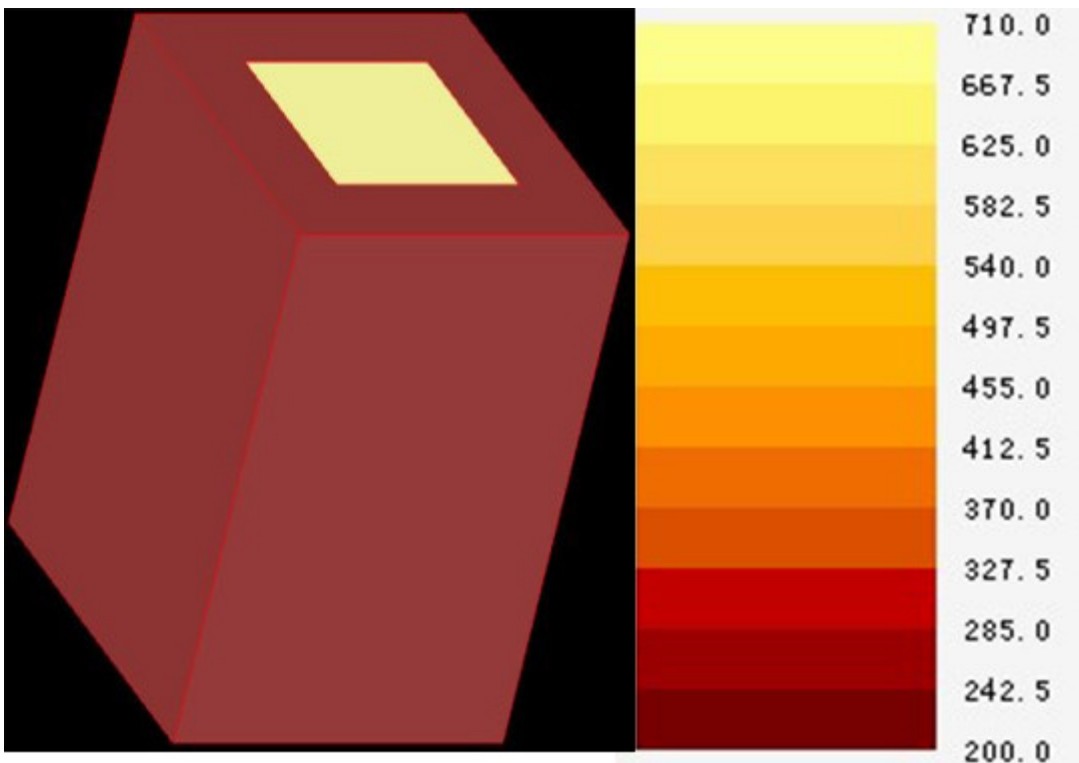

**Figure 2.** Original temperature of the mold and melt during simulation.

2.2.2. Parameter Setting

In the paper, the numerical simulations are based on the cavitation effect of the aluminum melt at 710 °C, which is influenced by the thermal property parameters. Table 1 provides the relevant thermal property parameters at 710 °C, as derived from Procast.

**Table 1.** Thermophysical properties of ingot and mold at 710 °C.

| Thermal Properties | Al-5.0Cu | H13Steel |
|---|---|---|
| Density/kg·m$^{-3}$ | 2450 | 7500 |
| Thermal conductivity/J·Kg$^{-1}$·°C$^{-1}$ | 15.5 | 28.8 |
| Young's modulus/GPa | 63 | 210 |
| Poisson's ratio | 0.3 | 0.3 |

In the study of cavitation effects, it has been observed that the convergence of calculations is positively influenced when appropriate inlet and outlet pressures and gas–liquid phase change rates are chosen. Combined with the results obtained from the optimized process parameters in the experiment, the following parameter settings were employed in the simulation:

The hydrogen content during the melting process needs to be controlled and therefore was set at 3 mL/kg, while the gas phase fractions at the inlet and outlet were adjusted to be a consistent value.

In this simulation, the involved relaxation factors have little effect on the momentum equation, and the specific relaxation factors need to be adjusted in the simulation according to the actual iteration steps and convergence for improved accuracy.

## 3. Experimental Factors, Materials, and Machines

Firstly, 8 kg of the Al-Cu alloy was smelted in a SG-7.5–12 well-type resistance furnace. Next, 0.5% of a commercial refining agent was introduced to refine and degas the melt. Then, after waiting for 5 min, the molten metal surface's impurities were removed, and the melt was subsequently poured into a preheated mold, which was preheated to 200 °C

in advance. In addition, the casting temperature needs to be controlled at 700~710 °C. Following the pouring of the molten alloy into the mold, the hydraulic press was promptly activated to apply pressure (25 MPa and 100 MPa). In addition, once the ultrasonic indenter contacted the metal melt, the ultrasonic device started to operate (500 W and 1 kW), ensuring the synchronous application of ultrasound and pressure.

The experiment was conducted on a vertical, four-column hydraulic press with a maximum load capacity of 100 kN. KISTLER 9130BQ05SP0 and NiCr-NiSi thermocouples were employed to monitor and collect the actual temperature and pressure data. The mold used in this experiment was made of H13 steel. In order to facilitate demolding, graphite oil was applied to the wall of the mold before starting the melting process. Figure 3 shows the three-dimensional schematic of the apparatus. When the temperature was preheated to 600 °C, the variable amplitude rod was horizontally immersed into the mold to a depth of 10 mm, resulting in the final dimensions of the ingot being Φ60 mm × 70 mm. Additionally, to provide a comprehensive comparison between the parts processed under different conditions and to understand the impact of coupled fields on their microstructures, some other samples were also prepared under the conditions of gravity casting, single extrusion pressure, and ultrasonic casting.

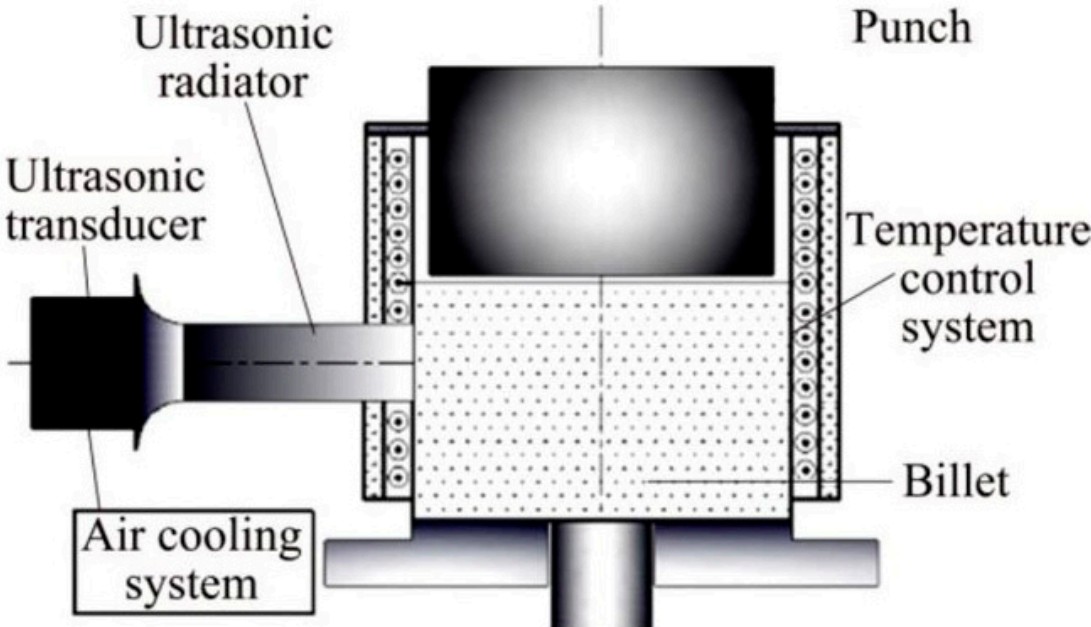

**Figure 3.** The combined ultrasonic vibration and applied pressure apparatus. The obtained sample was cut into a size of Φ12 mm × 10 mm. Given that the effects of cavitation and acoustic flow under the ultrasonic vibration are more significant in the local area near the end face of the variable amplitude rod [20], the metallographic samples were removed from position A, as shown in Figure 1. In this way, the performance of the process under different coupling parameters could be better studied. After the polished samples were corroded by 0.5% HF aqueous solution for 30 s, the microstructure was observed and analyzed by LEICA/DMI 5000 M metallographic microscope.

## 4. Results and Discussion

### 4.1. Numerical Simulation of the Coupled Field Cavitation

According to previous studies, the best static continuous vibration effect is achieved when an appropriate power is selected [21,22]. Consequently, the experimental parameters, shown in Figure 4, were adopted as a reference for our investigation of the coupling field. From the previous study, it can be seen that the closure time t of the bubble under the cavitation effect is as follows [13]:

$$t = 0.915 R_0 \left( \frac{\rho_0}{P_0 + P_n} \right)^{\frac{1}{2}} \tag{11}$$

where $R_0$ represents the initial radius of the bubble, $\rho_0$ denotes the density of the melt, $P_0$ refers to the static pressure acting on the melt, and $P_n$ is the internal pressure of the melt. As the pressure increases, the time t decreases accordingly. This indicates that once the number of bubble closures in a vibration period in the coupling field increases, the volume fraction of cavitation bubbles also increases. Figure 4 is the curve of the volume fraction of cavitation bubbles at the melt near the horn melt A with time. It can be seen that the curves under different parameters all reach the initial cavitation bubble volume fraction (1%) at the eighth time step, marking the beginning of the cavitation effect. Next, the volume fraction under different parameters increases rapidly with time, and several processes of growth–oscillation–collapse are completed at different times, and finally, enter the steady-state stage. Comparing the volume fraction peaks of curves (a) and (b), it can be found that the ultrasonic power is constant, and the two curves under different pressures enter the steady-state oscillation stage after 1200 μs. At this time, the maximum volume fraction of the cavitation bubble near the collapse increases. This shows that increasing the pressure in the coupling field can improve the cavitation intensity to a certain extent, but it has little effect on the time to enter the steady-state cavitation stage. In the curves (c) and (d), it can be found that when the ultrasonic power is 1 kW, increasing the pressure achieves a similar rule. When the pressure is constant and the ultrasonic power is increased, it can be seen that under 500 W power ultrasound, seven growth–collapses are completed within 1600 μs, while ten times are completed under 1 kW, which further proves that under the coupling effect, increasing the ultrasonic power parameter not only increases the number of cavitation effects in the melt per unit time, but it also increases the intensity of the cavitation effect.

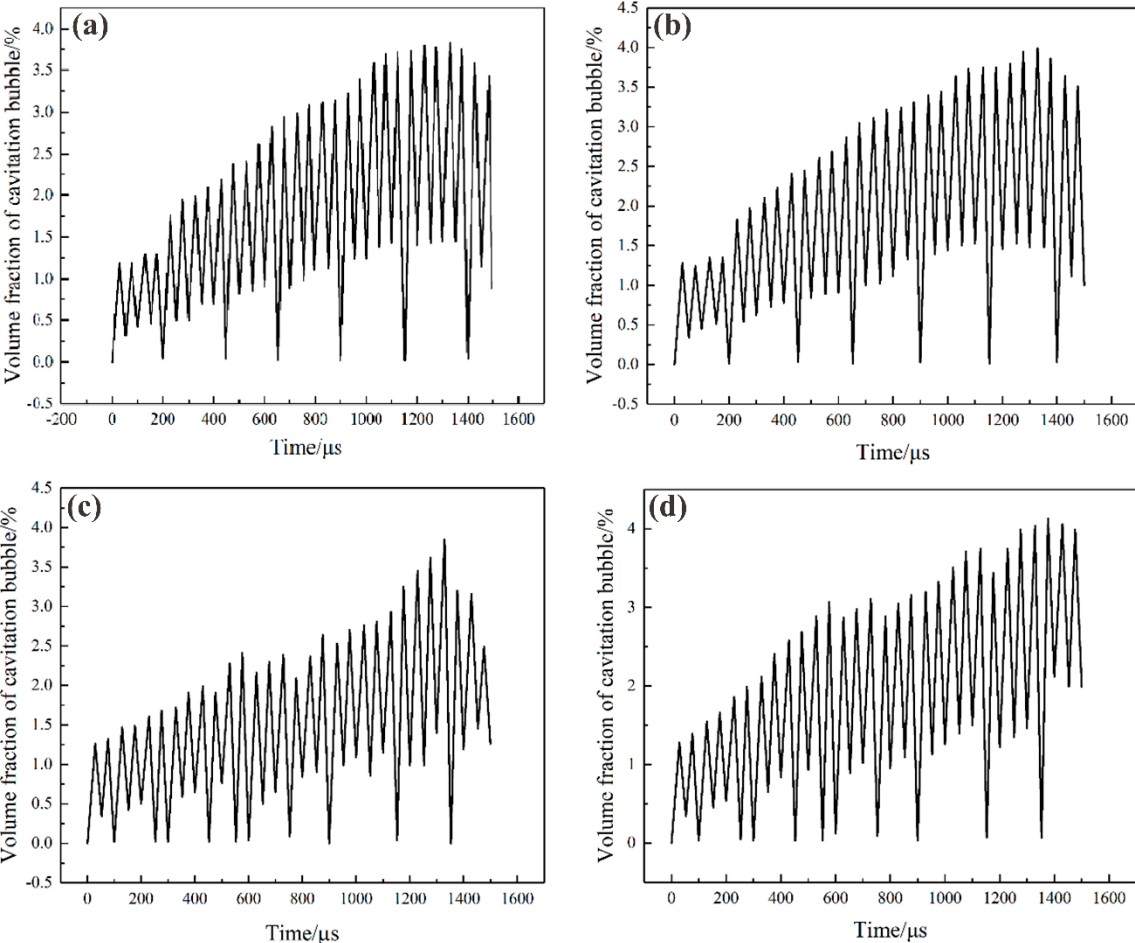

**Figure 4.** Curves of volume fraction of cavitation bubbles. (**a**) 25 MPa–500 W. (**b**) 100 MPa–500 W. (**c**) 25 MPa–1 kW. (**d**)100 MPa–1 kW.

The above results show that when in the ultrasonic–pressure coupling, with the increase in the pressure, the corresponding pressure in the bubble increases further, and the radius of the bubble collapses is also larger. With the increase in the ultrasonic power, the amplitude of the end face of the horn also increases, which can improve the vibration characteristics of the horn in the metal melt to a certain extent, which is reflected in the shortening of the time required for the cavitation bubble to grow and close.

Figure 5 illustrates the cloud map of pressure distribution obtained after the 400th extraction iteration. By comparing Figures 5a and 5b, it is evident that the positive pressure peak at the interface between the ultrasonic amplitude variation rod and the molten medium increased from 0.45 MPa to 0.99 MPa, while the negative pressure rose from 0.13 MPa to 0.34 MPa. This is because the temperature and pressure generated due to the collapse of the cavitation bubbles could reach to $6.55 \times 10^4$ K and $1.89 \times 10^3$ MPa, respectively. The strong shock wave causes a fluctuation of energy in the melt, leading to the formation of larger convective regions. Notably, the stirring that occurred near the variable-amplitude rod exhibited the most pronounced intensity in this phenomenon. A similar trend could also be observed when studying the effect of ultrasound power. However, because the energy of the ultrasonic wave attenuates along the propagation direction from its source, the influence of pressure parameters was more significant. According to Figure 5d, it could be found that the largest negative pressure area was formed at the end face of the horn, with peak positive and negative pressures of 1.81 MPa and 0.98 MPa, respectively. During the simulation process, the alternating stresses generated by the high and low pressure inside the melt would contribute to the strong convection.

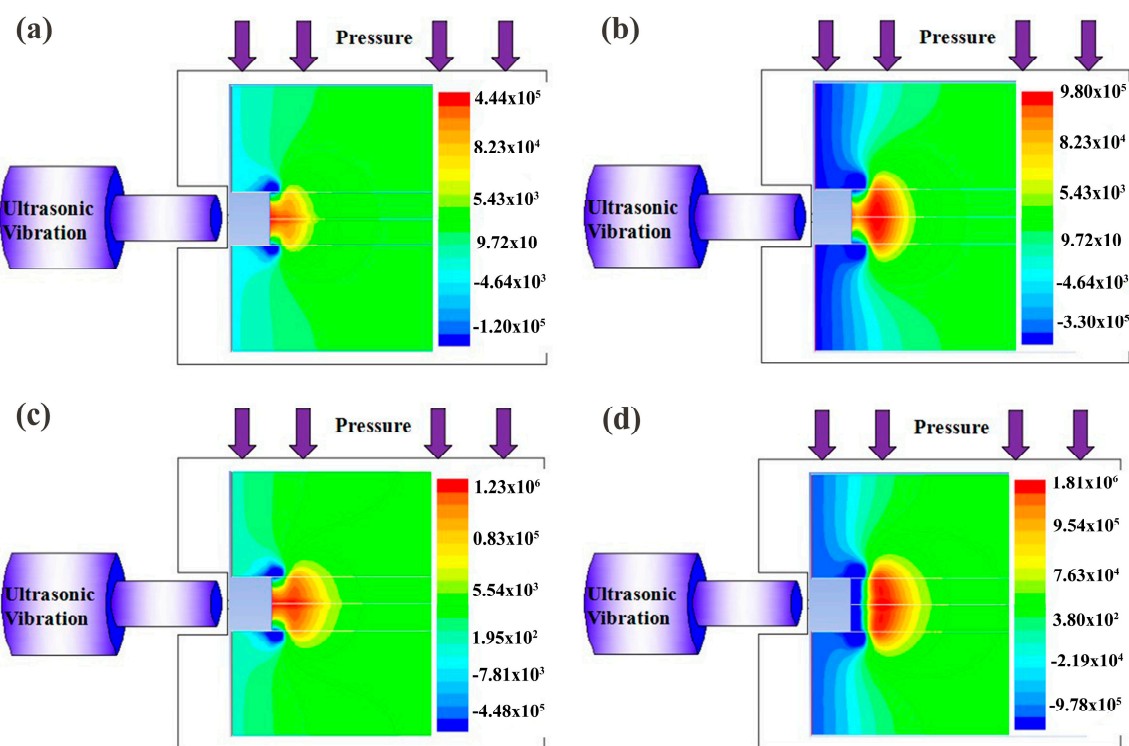

**Figure 5.** Distribution of inner pressure at 400th time step (Pa). (**a**) 25 MPa–500 W. (**b**) 100 MPa–500 W. (**c**) 25 MPa–1 kW. (**d**) 100 MPa–1 kW.

Figure 6 is the velocity distribution cloud map that is generated when the extraction iteration reaches the 500th step. The simulation revealed that the velocity field inside the melt is more sensitive to the pressure coefficient. As the pressure value increased, the area where the flow occurred became larger. The same pattern existed for the ultrasonic parameters. However, the flow pattern formed around the horn under the influence of pressure parameters was more complex. Small-scale impact zones were prone to form,

enhancing the stirring capability. The increased stirring capability makes the breakage of the dendrites easier. This observation is consistent with the inferences derived from the above pressure cloud diagram. When the pressure and ultrasonic parameters were adjusted to 100 MPa and 1 kW, the peak velocity at the junction of the melt and the amplitude rod reached 7.47 m/s, which is the highest level amongst these four cases.

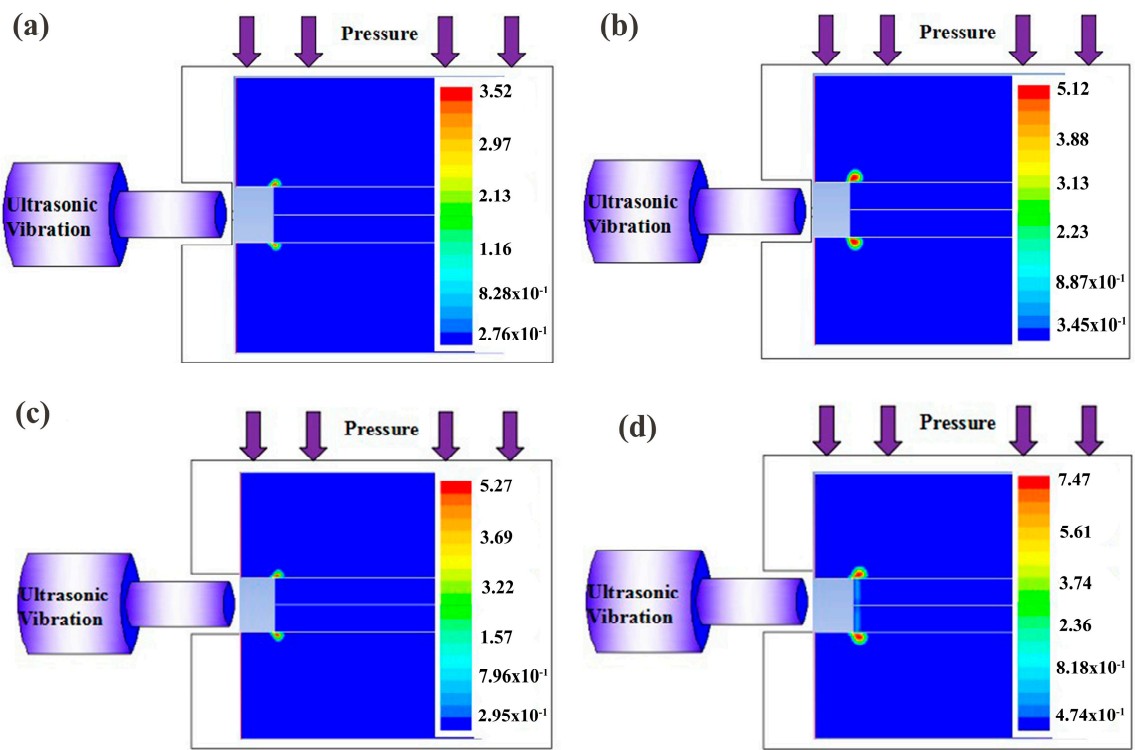

**Figure 6.** Distribution of inner velocity at 500th time step (m/s). (**a**) 25 MPa–500 W. (**b**) 100 MPa–500 W. (**c**) 25 MPa–1 kW. (**d**) 100 MPa–1 kW.

### 4.2. Experimental Verification

Figure 7 and Table 2 present the macroscopic corrosion and grain size of the cross-sections of parts processed under various coupling process parameters, respectively. A comparison between Figures 8e and 8f shows that, when the applied pressure increased from 25 MPa to 100 MPa, the average grain size decreased from 187.5 μm to 125.5 μm. However, it is noteworthy that the increase in pressure led to a consequential elevation in the proportion of columnar grains, ascending from 4.14% to 10.13%. Under the action of extrusion pressure, the aluminum alloy melts at about 710 °C and is closely spread on the mold wall, and the heat transfer is expected to be rapid. In addition, despite the grains growing rapidly into the slender columnar crystals along the direction of the heat transfer, the coupling effect promoted the formation of greater undercooling levels, resulting in an increased nucleation rate and a reduction in grain size. Subsequently, when the power was increased to 1 kW, it was found that the increased power enhanced the sound pressure gradient, changing the direction of single heat transfer and diffusion. Finally, the proportion of columnar crystals was suppressed.

**Table 2.** Analysis of the microstructure of Al-5.0Cu alloy at position A.

| Process Parameters | Average Grain Size/μm | Proportion of Columnar Crystal/% |
|---|---|---|
| 25 MPa and 500 W | 187.5 | 4.14 |
| 100 MPa and 500 W | 125.5 | 10.33 |
| 25 MPa and 1 kW | 134.4 | 3.15 |
| 100 MPa and 1 kW | 87.6 | 1.76 |

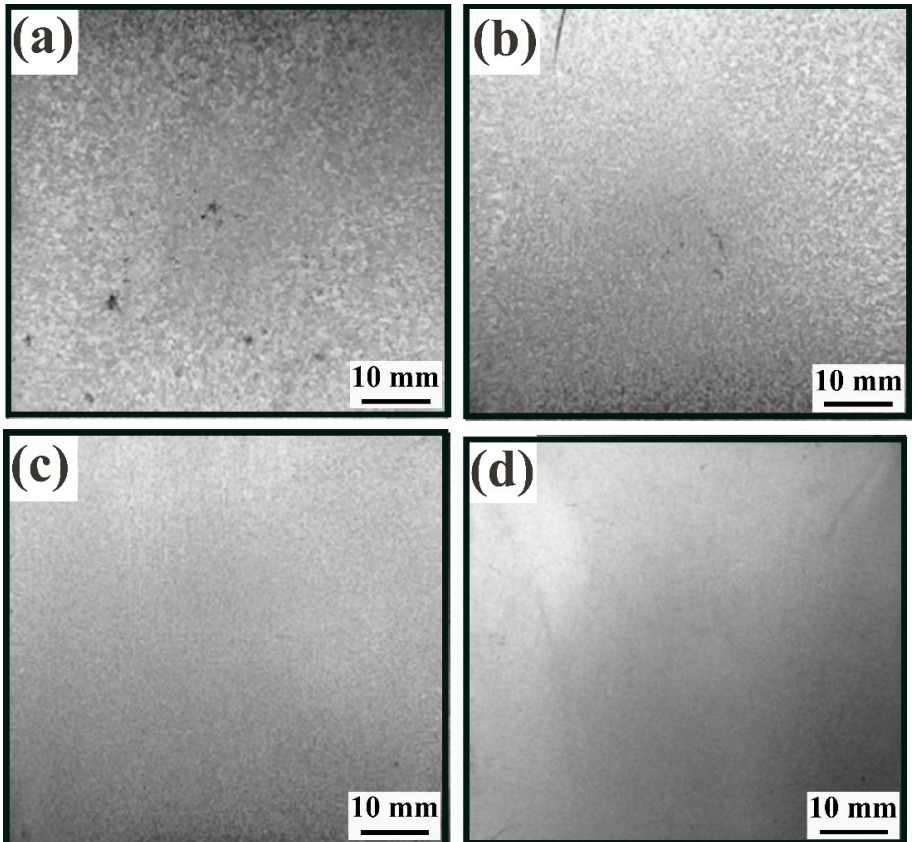

**Figure 7.** Solidification microstructure of the Al-5.0Cu alloy. (**a**) 25 MPa–500 W. (**b**) 100 MPa–500 W. (**c**) 25 MPa–1 kW. (**d**) 100 MPa–1 kW.

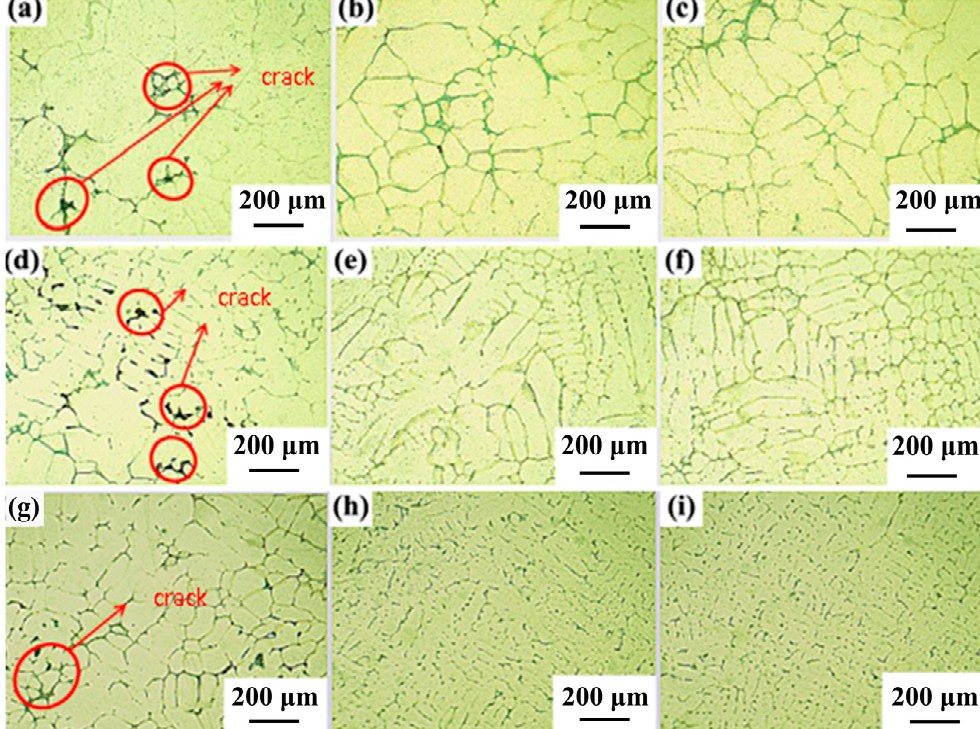

**Figure 8.** Microstructures of Al-5.0Cu alloy processed under different processing parameters. (**a**) Gravity field casting. (**b**) 25 MPa. (**c**) 100 MPa. (**d**) 500 W. (**e**) 500 W–25 MPa. (**f**) 500 W–100 MPa. (**g**) 1 kW. (**h**) 1 kW–25 MPa. (**i**) 1 kW–100 MPa.

For the ultrasonic power, the obtained results show intuitively that the increase in its value has an inhibitory effect on both the average grain size and the growth of columnar grains. In the ultrasonic field, the mechanical energy generated by the crystal resonance could break the initial dendrite chains, causing the fracture of the dendrite arm root of the tree-like crystals and their re-melting into the liquid phase. This process increases the driving force for phase transformation and enhances the refinement effect on the microstructure of solidified parts, promoting the uniform distribution of solute elements. As depicted in Figure 8d, when simultaneously setting the ultrasonic power and extrusion pressure to 100 MPa and 1 kW, the obtained average grain size and the proportion of columnar grains exhibited an ideal outcome.

In summary, the increase in both ultrasonic power and extrusion pressure led to a decrease in the average grain size. The impact of the pressure parameter was, in this regard, more significant than the power. The increase in pressure also corresponds to an enhancement in the proportion of columnar grains. However, this is not the case for the ultrasonic power. Consequently, when optimizing the process parameters, the experimental results achieved using 1 kW of ultrasonic power were more consistent with expectations.

Figure 8 shows the microstructure of the parts processed under different process parameters. There were many hole defects in the parts processed by the gravity casting field (Figure 8a). In the parts processed by the pressure field, smaller grain sizes were observed, and the casting defects were reduced compared to those of the gravity casting field parts. However, a higher proportion of fine and elongated dendritic structures was present in these parts. Due to the unique physical and chemical reactions caused by the ultrasonic vibration, the obtained microstructure mostly consisted of equiaxed crystals. However, conventional casting defects were not completely eliminated by the ultrasonic vibration (Figure 8d,g). Considering the limited effect of a single field on the grains and defects, the micro-morphologies of the parts processed under different combinations of 500 W ultrasonic power and extrusion pressure were compared. The results show that the secondary dendrite spacing (SDAS) of the microstructure of parts processed under the coupling field was reduced (Figure 8e,f). Furthermore, by increasing the ultrasonic power to 1 kW and coupling it with the corresponding extrusion pressure once again, the dendrite spheroidization phenomenon occurred. Moreover, a uniform, rounded microstructure was observed at the interface, along with a finer and more uniformly distributed $\theta$ phase in the alloy when the extrusion pressure reached 100 MPa. This is primarily attributed to the synergistic effect of ultrasound and extrusion, which accelerates solute diffusion in the melt and promotes the generation and detachment of more nucleation sites. Moreover, the dendritic skeleton structure is more easily dispersed by the liquid phase flow, resulting in more uniform nuclei dispersion and weaker macrosegregation. Hence, by optimizing the parameters of ultrasound power and extrusion pressure, a fine and evenly distributed grain structure could be obtained in the processed parts.

## 5. Conclusions

This study employed FLUENT 14.0 to conduct numerical simulations of the coupled effects of ultrasound and pressure on the process optimization of Al-Cu alloys. Subsequently, the experimental platform for ultrasonic–pressure coupling casting aluminum alloy melt was built, and the coupling parameters were designed and optimized according to the requirements of the coupling process. Finally, the purpose of improving performance was achieved. The conclusions obtained are as follows:

(1)	Ultrasonic–pressure coupling could enhance the acoustic flow and cavitation effects in the aluminum melt. Through the numerical simulation of the cavitation effect under different pressure and ultrasonic power parameters, it was found that increasing the pressure, while enlarging the area of cavitation and molten convection, shortens the period of cavitation bubble growth–collapse. As for the ultrasound power, its increase, as well as promoting higher melt flow velocities, significantly affected the peak values of positive and negative pressures within the melt.

(2)    By comparing the effect of different process parameters on the alloy structure, it was observed that a single-pressure field could basically eliminate casting defects, such as holes existing in the microstructure, and refine the secondary dendrite spacing. However, it was difficult to obtain fine and uniformly equiaxed crystal structure parts. Conversely, a single ultrasonic field, despite being able to lead to uniform equiaxial crystal structures, could not eliminate the above defects.

(3)    When the applied pressure increased from 25 MPa to 100 MPa, the average grain size decreased from 96.7 μm to 71.4 μm. The coupled field of ultrasound and pressure (100 MPa and 1 kW) could further reduce the grain size and decrease the proportion of columnar grain structures. Additionally, it enhanced the uniformity of microscopic tissue distribution at the sampling location, facilitating the attainment of parts with a fine and uniformly equiaxed grain structure.

(4)    In the microscopic numerical simulation, the influence of extrusion force on the macroscopic temperature field and the influence of power ultrasound on the nucleation rate and growth coefficient are fully considered. However, there is a lack of an effective calculation model to add the ultrasonic–extrusion coupling effect to the simulation of the macroscopic temperature field, which needs to be continuously improved and improved in the follow-up.

**Author Contributions:** Conceptualization, J.W. and B.L.; methodology, J.W.; software, K.K.; validation, Y.Z., J.W. and Y.C.; formal analysis, W.Z.; investigation, J.W.; resources, B.L.; data curation, Y.C.; writing—original draft preparation, J.W.; writing—review and editing, K.K.; visualization, Y.Z.; supervision, B.L.; project administration, B.L.; funding acquisition, B.L. All authors have read and agreed to the published version of the manuscript.

**Funding:** This study was funded by the National Natural Science Foundation of China (52265043), Guizhou Province outstanding scientific and technological talents project (YQK(2023)011, Guizhou Provincial Science and Technology Projects (ZK2023(014)).

**Data Availability Statement:** The data presented in this study are available on request from the corresponding author.

**Conflicts of Interest:** The authors declare no conflict of interest.

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
