# Peer review of "Numerical/Experimental Study of Process Optimization Conducted on an Al-Cu Alloy Produced by Combined Fields of Applied Pressure and Ultrasonic Vibration"

_crystals, doi:10.3390/cryst13101466_

Round 1
Reviewer 1 Report
Authors analyzed the cavitation and acoustic streaming effects characteristics of an Al-Cu alloy treated under different coupling process parameters, related to applied extrusion pressure and ultrasonic vibration by combining numerical simulation and experimental verification. This research is characterized by clarity of purpose and methodology, as well as soundness of results, method of presentation, depth of discussion, and integrity of conclusions.
Therefore I strongly recommend the acceptance of this paper for publication but after taking into consideration the following comments:-
1- Please distinguish the contribution of the second author because he is from the department of genetic, chemical, and biotechnology which places him beyond the scope of this research.
2- In the abstract section, address the experimental technique and the applicability of this work in more detail. In addition, the abstract should be concise and summarize the main findings
3- Rewrite the introduction to give a consistent literature review. Answering the following questions may assist the authors: Why are these works significant? What specific issues were addressed? How do earlier results connect to the most recent work? What are the current unresolved research issues? Answering the questions naturally leads to the novelty of the suggested work.
4- Indicate whether the formulas used are from a reference, standard equations such as the Helmholtz equation, or were created by the authors.
5- Please explain in line 95 whether the symbol is Q1 or r1 because it is unclear to readers.
6- Please add figure 4 to figure 1 because it explains the geometric model better.
7- Although the results are adequately presented, they still require in-depth study and comparison with similar findings, therefore please conduct more research and compare your findings with similar ones to demonstrate the novelty of your work.
8- The data should be used to back up the conclusions further.
9- Please try to use more references because the present 17 are insufficient to support work such as this research.
10- Include a brief recommendation for future work based on your findings and conclusions.
Author Response
(1) Please distinguish the contribution of the second author because he is from the department of genetic, chemical, and biotechnology which places him beyond the scope of this research.
- Regarding the contribution of the second author, we have made corresponding modifications. Although he comes from other fields, he does make a lot of contributions in this study.
(2) In the abstract section, address the experimental technique and the applicability of this work in more detail. In addition, the abstract should be concise and summarize the main findings.
- Thanks for the referee’s suggestion and remind, We have modified the abstract part accordingly.
(3) Rewrite the introduction to give a consistent literature review. Answering the following questions may assist the authors: Why are these works significant? What specific issues were addressed? How do earlier results connect to the most recent work? What are the current unresolved research issues? Answering the questions naturally leads to the novelty of the suggested work.
- Thank you for your careful work. We have modified the introduction part accordingly.
(4) Indicate whether the formulas used are from a reference, standard equations such as the Helmholtz equation, or were created by the authors.
- Thank for referee’s valuable and thoughtful suggestion. We have annotated the references for the formulas that appear.
(5) Please explain in line 95 whether the symbol is Q1 or r1 because it is unclear to readers.
- Thank for referee’s valuable and thoughtful suggestion. I 'm sorry, we checked several times and found no Q1 or r1
(6) Please add figure 4 to figure 1 because it explains the geometric model better.
- Thank for referee’s valuable and thoughtful suggestion. We have improved Figure 1 and Figure 4.
(7) Although the results are adequately presented, they still require in-depth study and comparison with similar findings, therefore please conduct more research and compare your findings with similar ones to demonstrate the novelty of your work.
- Thank for your valuable and thoughtful suggestion. The main research object of this paper is the ultrasonic vibration system, and there are few studies on the ultrasonic power supply. Therefore, it is necessary to further study and design the ultrasonic power supply with low sensitivity to load changes.
(8) The data should be used to back up the conclusions further.
- Thank for your valuable and thoughtful suggestion. Combined with the opinions of another reviewer, we revised the conclusion.
(9) Please try to use more references because the present 17 are insufficient to support work such as this research.
- Thank for your valuable and thoughtful suggestion. We have enriched the literature.
(10) Include a brief recommendation for future work based on your findings and conclusions.
- Thank for your valuable and thoughtful suggestion. Our outlook for future work is as follows.

Reviewer 2 Report
Dear Author(s), please find below comments raised on the manuscript titled ‘Numerical-Experimental Study of Process Optimization Conducted on an Al-Cu Alloy Produced by Combined Fields of Applied Pressure and Ultrasonic Vibration‘, Manuscript ID: crystals-2583630.
The most important comments are below:
1. In the ‘Abstract’ section, some words on the area of study were not introduced. This section should contain, beginning with the field of analysis, the proposal, and some concluding sentences with one general advantage. Limitations of the study proposed would be welcome as well.
2. For the ‘Introduction’ section, the sentences in lines 30-68 do not include some critical review. The only advantages of previous studies are presented. In the current form, the gaps from the mentioned lines and the last, lines 69-77, are not proper and the proposal (novelty) does not derive from the lack in the current state of knowledge. Both parts look like separate sections.
3. Reviewing the section 2.1, the Helmholtz equation (1) is not newly proposed. In that case should be referenced to the primary sources. Similar to all of the equations proposed, if are not newly proposed by the Author(s), must be cited with the first sources. Ony the equation (14) was properly referenced.
4. Some values of the experiment in section 2.2.1 are not obvious. For example, the vibration frequency was 20 kHz, why? Some of the values look like selected arbitrarily (more examples: variable amplitude rod amplitude of A=10 μm, etc.). The Author(s) should add some justifications or references to the previous studies.
5. As in the previous comment, similar, not explained, values in section 3 are raised. It must be presented more comprehensively, giving the reader into full understanding of what proposal are Author(s) trying to convey.
6. Improving section 3, the Author(s) should propose the flow chart of the experiment. In the current form, it is difficult how many crucial factors can influence the final results.
7. In the ‘Experimental verification’ section, no. 4.2, there is no discussion. In fact, it is difficult to resolve what is the main advantage of the novelty and study proposed. Moreover, some limitations of the experiment should be raised including some further prospects of the study.
8. Similar to the previous issue, any further proposals should be raised. In the current form, the Author(s) did mention what can be the future motivation in the experiment. The main advantages were not appropriately emphasized.
9. In the ‘Conclusion’ section there is no main purpose highlighted. The second conclusion should be divided into two or three separate sentences. There are too many ideas presented in one gap so the Author(s) divide it.
10. All of the cited references should include full DOI links to improve fast recognition by the reader. Moreover, they should be formatted according to the journal template requirements.
Further, some additional, editorial issues, must be raised, as below:
11. The quality of the descriptions (text) on some figures, like Figures 1 and 5, must be improved.
12. There are many parameters and shortcuts in the body text of the manuscript so, consequently, the ‘Nomenclature’ section should be provided.
13. There is a double space in line 287.
From all of the above comments, the manuscript should be revised.
Author Response
Reviewer 2
(1) In the ‘Abstract’ section, some words on the area of study were not introduced. This section should contain, beginning with the field of analysis, the proposal, and some concluding sentences with one general advantage. Limitations of the study proposed would be welcome as well.
- Thank for referee’s valuable and thoughtful suggestion. We have made corresponding improvement.
(2) For the ‘Introduction’ section, the sentences in lines 30-68 do not include some critical review. The only advantages of previous studies are presented. In the current form, the gaps from the mentioned lines and the last, lines 69-77, are not proper and the proposal (novelty) does not derive from the lack in the current state of knowledge. Both parts look like separate sections.
- Thank for referee’s valuable and thoughtful suggestion. We have rewritten the introduction.
(3) Reviewing the section 2.1, the Helmholtz equation (1) is not newly proposed. In that case should be referenced to the primary sources. Similar to all of the equations proposed, if are not newly proposed by the Author(s), must be cited with the first sources. Only the equation (14) was properly referenced.
- Thank for referee’s valuable and thoughtful suggestion. We have annotated the references for the formulas that appear.
(4) Some values of the experiment in section 2.2.1 are not obvious. For example, the vibration frequency was 20 kHz, why? Some of the values look like selected arbitrarily (more examples: variable amplitude rod amplitude of A=10 μm, etc.). The Author(s) should add some justifications or references to the previous studies.
- Thank for referee’s valuable and thoughtful suggestion. Ultrasonic vibration with a frequency of 20 kHz has a moderate penetration depth, higher mechanical power and lower production cost, making it an ideal choice in many application fields. We also cited references. In addition, the simulation model value should be combined with the actual model. The size of the experimental part is as follows.
(5) As in the previous comment, similar, not explained, values in section 3 are raised. It must be presented more comprehensively, giving the reader into full understanding of what proposal are Author(s) trying to convey.
- Thank for referee’s valuable and thoughtful suggestion. The selection of pressure and ultrasound is based on the introduction and the previous single-field research results of our research group.
(6) Improving section 3, the Author(s) should propose the flow chart of the experiment. In the current form, it is difficult how many crucial factors can influence the final results.
- Thank for referee’s valuable and thoughtful suggestion. We have modified the experimental flow chart. The pressure, power and temperature values are monitored throughout the process.
(7) In the ‘Experimental verification’ section, no. 4.2, there is no discussion. In fact, it is difficult to resolve what is the main advantage of the novelty and study proposed. Moreover, some limitations of the experiment should be raised including some further prospects of the study.
- Thank for referee’s valuable and thoughtful suggestion. In this part, we have made a discussion, and the conclusion has also been improved.
(8) Similar to the previous issue, any further proposals should be raised. In the current form, the Author(s) did mention what can be the future motivation in the experiment. The main advantages were not appropriately emphasized.
- Thank for referee’s valuable and thoughtful suggestion. In the introduction, we emphasize this point, and the results also confirm it.
(9) In the ‘Conclusion’ section there is no main purpose highlighted. The second conclusion should be divided into two or three separate sentences. There are too many ideas presented in one gap so the Author(s) divide it.
- Thank for referee’s valuable and thoughtful suggestion. We have made corresponding improvements to the conclusion.
(10) All of the cited references should include full DOI links to improve fast recognition by the reader. Moreover, they should be formatted according to the journal template requirements.
- Thank for referee’s valuable and thoughtful suggestion. We have improved the quality of references.
(11) The quality of the descriptions (text) on some figures, like Figures 1 and 5, must be improved.
- Thank for referee’s valuable and thoughtful suggestion. We have improved the quality of the text description of Figure 1 and Figure 4. Due to the merging of the figure, the original Figure 5 becomes Figure 4.
(12) There are many parameters and shortcuts in the body text of the manuscript so, consequently, the ‘Nomenclature’ section should be provided.
- Thank for referee’s valuable and thoughtful suggestion. Due to the large number of parameters used in the study, it is not meaningful to provide specific names for each parameter. Improve individual.
(13) There is a double space in line 287.
- Thank for referee’s valuable and thoughtful suggestion. By searching, we correct the double spaces in the text.

Reviewer 3 Report
I don't think the citation of bibliographic references used by MDPI journals is respected, please check.
Figure 1 is too large.
More explanation is needed about the data acquisition system (temperature and pressure).
Figure 3 is also too large. Instead of a very sketchy 3D assembly, a block diagram of the experimental layout might be more useful.
What is the purpose of Figure 4 and why is it so large?
Figure 5 has very poor resolution.
Minor editing of English language required.
Author Response
Reviewer 3
(1) I don't think the citation of bibliographic references used by MDPI journals is respected, please check.
- Thank for referee’s valuable and thoughtful suggestion. We modify the references.
(2) Figure 1 is too large.
- Thank for referee’s valuable and thoughtful suggestion. We have modified Figure 1.
(3) More explanation is needed about the data acquisition system (temperature and pressure).
- Thank for referee’s valuable and thoughtful suggestion. The pressure is controlled by the press console, and the temperature is measured as shown. Thank you again for your careful work.
(4) Figure 3 is also too large. Instead of a very sketchy 3D assembly, a block diagram of the experimental layout might be more useful.
- Thank for referee’s valuable and thoughtful suggestion. We have optimized the graph.
(5) What is the purpose of Figure 4 and why is it so large?
- Thank for referee’s valuable and thoughtful suggestion. Because the sample tissues at different locations may be different, Figure 4 wants to express the location of our sampling. Combined with another reviewer 's opinion, we have put Figure 4 together with Figure 1.
(6) Figure 5 has very poor resolution.
- Thank for referee’s valuable and thoughtful suggestion. Based on the original image, we modified it.

Reviewer 4 Report
The study is aimed at improving the quality of aluminum alloy castings by combining the effects of pressure and ultrasound. The first part of the paper is devoted to computer simulation of this process, including a description of mathematical models and boundary conditions, the second part is devoted to experimental verification of the results. The results are of scientific and practical interest. However, before the paper can be published, a number of problems should be solved in it.
1. What determines the choice of a specific Al-Cu alloy for this research?
2. Can the authors say to what extent the results can be extended to other aluminum alloys? In other words, how universal are the obtained results?
3. When analyzing the sizes and morphology of grains, the authors refer to Fig. 8. From this figure, it can be seen that with an increase in both pressure and power, the contrast in the structure weakens and the image becomes more uniform. This can indeed be associated with the structure refinement. However, individual grains (and even more so their shape) are not visible at all in these images. In addition, even the scale mark is not given. One could refer to Figure 9, however, there is a contradiction with the data in Table 2. The size of the grains in the Figure 9 (e, f, h, i) seems to be much smaller than presented in Table 2. This point needs to be explained in detail.
4. Lines 218-219. ‘…the volume fraction of cavitation bubbles increases from 3.99 % to 4.13 %’. This difference seems to be negligible.
Author Response
Reviewer 4
(1) What determines the choice of a specific Al-Cu alloy for this research?
- Thank for referee’s valuable and thoughtful suggestion. The introduction was modified to explain why the alloy was selected.
(2) Can the authors say to what extent the results can be extended to other aluminum alloys? In other words, how universal are the obtained results?
- The coupling field can regulate the crystal structure and macroscopic properties of Al-Cu alloys by changing the processing parameters and alloy composition. However, due to the differences in material properties and processing conditions, the coupling field effects in different alloy systems and specific application scenarios may be different. Therefore, in practical applications, experimental verification and parameter optimization are needed to obtain the best effect and universality.
(3) When analyzing the sizes and morphology of grains, the authors refer to Fig. 8. From this figure, it can be seen that with an increase in both pressure and power, the contrast in the structure weakens and the image becomes more uniform. This can indeed be associated with the structure refinement. However, individual grains (and even more so their shape) are not visible at all in these images. In addition, even the scale mark is not given. One could refer to Figure 9, however, there is a contradiction with the data in Table 2. The size of the grains in the Figure 9 (e, f, h, i) seems to be much smaller than presented in Table 2. This point needs to be explained in detail.
- We are sorry for our carelessness. The scale of the figure has been added. In addition, due to the error of text editing, the table and the figure are not matched, and we have also made corresponding modifications. Thank you again for your careful work.
(4) Lines 218-219. ‘…the volume fraction of cavitation bubbles increases from 3.99 % to 4.13 %’. This difference seems to be negligible.
- Thank for referee’s valuable and thoughtful suggestion. We improve the description.

Round 2
Reviewer 1 Report
The paper was adjusted and the manuscript is now acceptable for publishing because the authors considered the reviewers' comments and responded to them correctly and successfully.
Author Response
I sincerely thank you for your work.
Reviewer 2 Report
The submission was improved appropiately so can be considered for publication.
Author Response
I sincerely thank you for your work.Reviewer 3 Report
The authors have provided sound explanations and solutions for every issue pointed by the reviewer. In my opinion the paper can now be published in the journal.
Minor editing of English language required.
Author Response
Thank you for the referee's suggestions and reminders. We have made the necessary modifications accordingly.
Reviewer 4 Report
Minor edits required before final publication:
1) Please check the “mesh size” on the page 5: the mesh size is listed as 0.8 mm, but below 0.08 mm.
2) Page 7, second paragraph. Should be ‘when the temperature’ instead of ‘When the temperature’.
3) On the page 11 the authors refer to figure 7 e and f, but figure 7 consists only of parts a-d.
4) Page 14, top paragraph. Should be ‘the experimental’ instead of ‘The experimental’.
5) I don't recommend using abbreviations in Conclusion section (namely SDAS in conclusion 2).
Author Response
Response to reviewers
Dear editor and reviewer:
Thank you for your letter and for the reviewer’s comments concerning our manuscript entitled "Numerical-Experimental Study of Process Optimization Conducted on an Al-Cu Alloy Produced by Combined Fields of Applied Pressure and Ultrasonic Vibration". Your comments are valuable and helpful for revising and improving our manuscript, as well as the important guidance for our research. We have studied the comments carefully and have made corrections, which we hope will meet with approval. The responses to the reviewer’s comments are as follows:
Reviewer
(1) Please check the “mesh size” on the page 5: the mesh size is listed as 0.8 mm, but below 0.08 mm.
- Thanks for the referee’s suggestion and remind. We have made corresponding modification.
(2) Page 7, second paragraph. Should be ‘when the temperature’ instead of ‘When the temperature’.
- Thanks for the referee’s suggestion and remind. We have made corresponding modification.
(3) On the page 11 the authors refer to figure 7 e and f, but figure 7 consists only of parts a-d.
- Thank you for your careful work. We have made corresponding modification.
(4) Page 14, top paragraph. Should be ‘the experimental’ instead of ‘The experimental’.
- Thank you for your careful work. We have made corresponding modification.
(5) I don't recommend using abbreviations in Conclusion section (namely SDAS in conclusion 2).
- Thank you for your careful work. We have made corresponding modification.
Once again, thank you very much for your comments and suggestions. The manuscript has been resubmitted to your journal. We are eagerly awaiting your positive response.
Sincerely,
Bo Lin
